# On Boundary Conditions for Damage Openings in RoPax-Ship Survivability Computations

**Petri Valanto**

Hamburgische Schiffbau-Versuchsanstalt GmbH (HSVA), Bramfelder Strasse 164, D-22305 Hamburg, Germany; valanto@hsva.de

**Abstract:** The survivability of a damaged RoPax ship in the case of a flooding accident can be critical, as these ships have a tendency for a rapid capsize. Various simulation tools are presently in use to study the behavior of damaged RoPax and cruise ships. Recent benchmark tests show that the numerical tools for this purpose are very useful, but their accuracy and reliability still leave something to be desired. In many numerical simulation codes for ship survivability, the water inflow and outflow through a damage opening are modeled with Bernoulli equation, which describes steady flow in an inertial frame of reference. This equation takes neither the floodwater inertia in the opening into account nor does it regard the effect of ship motions on the flow in the opening. Thus, there are some approximations involved in the use of the Bernoulli equation for this purpose. Some alternative formulations are possible. This study sheds light on the question of how relevant is it to use the more complicated formulations instead of the very simple and robust Bernoulli model in the numerical simulation of damaged ships in the sea.

**Keywords:** ship damage stability; numerical simulation; Bernoulli equation; model tests

## 1. Introduction

The survivability of damaged RoPax ships in the event of a flooding accident can be critical, as these ships have a tendency for rapid capsize, often not allowing for an orderly evacuation. Various simulation tools have been developed to study the behavior of damaged RoPax and cruise ships. The flooding process on and the motions of damaged ships in the seaway are complex, interacting processes that are difficult to simulate accurately and reliably. The recent benchmark tests on RoPax by Ruponen et al. [1] and cruise ships by Ruponen et al. [2] show that, albeit the numerical tools for this purpose are very useful, their accuracy and reliability still leave something to be desired.

In many numerical simulation codes for ship survivability, the water inflow and outflow through a damage opening on a ship side or bottom are modeled with Bernoulli equation (BE). The BE describes a steady flow in an inertial frame of reference. This equation takes neither the floodwater inertia in the opening into account nor does it regard the effect of the prevailing flow direction in the opening. Beside this, the damaged ship floating in waves is not rigorously an inertial frame of reference. Thus, there are quite a few approximations involved in the fairly common use of the BE for this purpose. Lee [3] used the dynamic orifice equation (DOE) as an alternative. This raises the question of how important is it to use the more complicated DOE or something more advanced instead of the very simple and robust BE in the numerical simulation of damaged ships in the seaway.

The RoPax ship under this study is a modern northern RoPax design made for research purposes only. The safety of this design was studied by various partners in the framework of the EU project Flooding Accident Response (FLARE) at a few levels of sophistication in several damage cases. HSVA carried out forensic analysis on several damage cases on this ship design using HSVA Rolls [4–7] as a simulation program. All sea

states in this investigation were modeled with JONSWAP-Spectrum, with a peak enhancement factor γ of 3.3 and a peak wave period $T_P$ of 10.0 s.

The in-house version of the Rolls code, HSVA Rolls, was used in all computations. Floodwater in internal compartments and on decks can be modeled either with shallow-water equations (SWE) or with a pendulum model. For all cases in this study, SWEs were used to model the flow on the trailer deck, and the pendulum model was used for the more deeply flooded compartment spaces below. The flow rates through the breaches are based on Bernoulli's equation. For the ship heave, pitch, sway, and yaw motions, the method uses response amplitude operators (RAO) determined in the frequency domain with a linear strip method. The roll and surge motions are determined by the time integration of the non-linear equations of motion coupled with the other four degrees of freedom. The hydrodynamic contributions are based on linear strip theory and on those based on the water motions in internal compartments. The hydrostatic contributions in calm water and waves are non-linear and are based on calculations with NAPA software [8].

## 2. General Description of the RoPax Vessel

The ship used in the numerical simulations and model tests is a 162 m long RoPax vessel designed to the SOLAS 2020 standard by Meyer Turku (MT) shipyard. The ship is designed as a day ferry, hosting up to 1900 passengers and a crew of 91. It has 800 m trailer lanes on the main trailer deck at 9.2 m above baseline and 1050 m of car lanes in the garage deck [9]. The main particulars of the vessel at the test draught are given in Table 1, and views of the ship design and of the scale model in the tests are given in Figures 1 and 2.

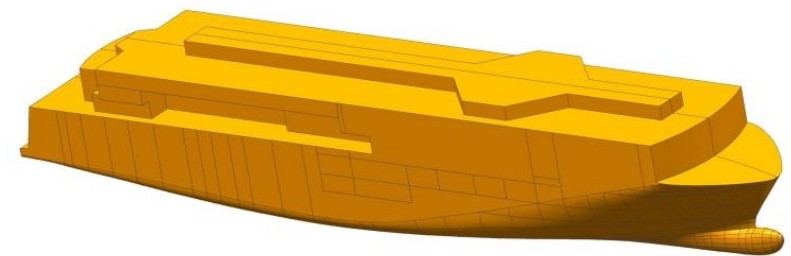

**Figure 1.** CAD-model of the MT RoPax used in the numerical simulations.

**Table 1.** Main data of the vessel in intact (test) condition.

| MT RoPax—HSVA Model no: 5460/5539 | Symbol | Unit | Ship |
|---|---|---|---|
| Length overall | $L_{OA}$ | m | 162.00 |
| Length between perpendiculars | $L_{PP}$ | m | 146.72 |
| Breadth at the waterline | $B_{WL}$ | m | 28.00 |
| Draught at the aft perpendicular | $T_A$ | m | 6.10/6.30 |
| Draught at the forward perpendicular | $T_F$ | m | 6.10/6.30 |
| Depth to trailer deck | D | m | 9.20 |
| Displaced volume (bare hull) | $\nabla_{BH}$ | m³ | 16,799.4 |
| Block coefficient | $C_B$ | - | 0.6522 |
| Intact transverse GM | GM | m | 1.425–3.40 |

Two versions of the MT RoPax design with minor differences in the subdivision were used in the numerical simulations for comparison with (1) FLARE Benchmark test experimental data and (2) FLARE Flooding Mitigation test experimental data, as described in Sections 5 and 6. The ship subdivision for the first case is described in [1], and for the second in [8].

### 3. Observations from Model Tests and Simulations

*3.1. Introduction*

The experience with the simulation code HSVA Rolls has shown that the computed results in beam seas tend to be slightly conservative in comparison with model test results: Thus, the computed times to capsize (TTC) are in general shorter than the TTCs obtained from model tests in beam seas. Similarly, the ship survives in model tests at higher significant wave heights than it does in the numerical simulations. The difference between the computations and model test results in terms of the significant wave heights related to the capsize boundary in beam seas can be as low as 0.5 m, but in some unfavorable cases it can also be higher. So far, no case is known in which the computed results would have given an essentially higher survivability than the corresponding model tests.

In view of this often encountered difference between the computations and model test results, the flow through a damage opening in a ship in calm water and in waves was investigated in HSVA and in the framework of the EU-funded research project Flooding Accident Response (FLARE). Figure 2 shows a damaged RoPax ship model tested in irregular beam seas during the FLARE model test campaign. The damage opening can be seen on the starboard, wave side of the vessel. At the opening, there are sensors for the measurement of flow speed and water elevation on the vehicle deck.

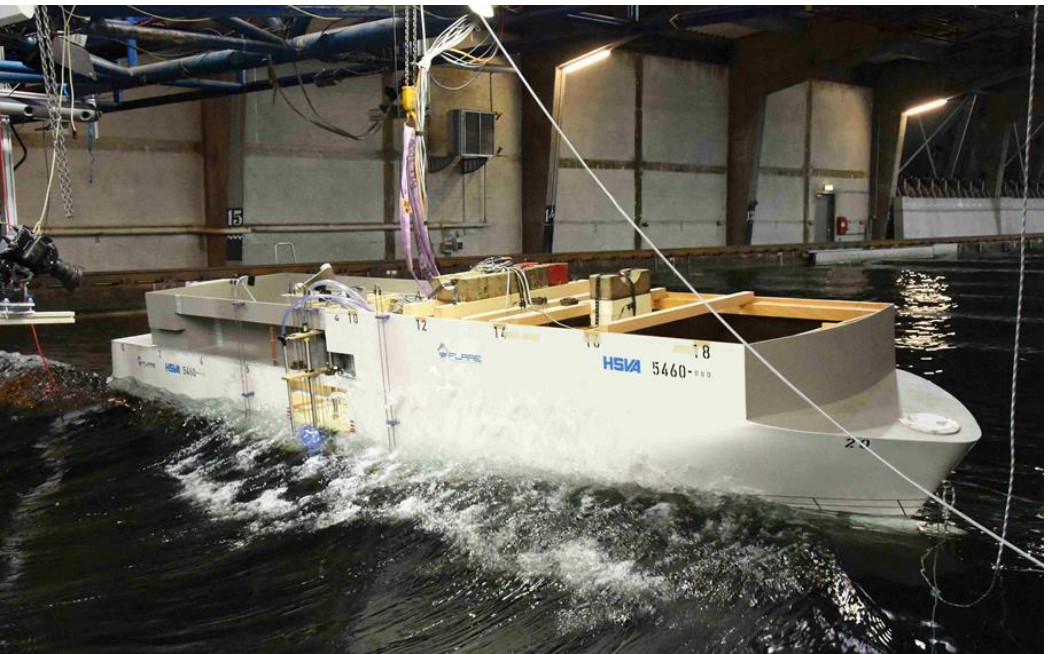

**Figure 2.** A model of a damaged RoPax ship in irregular beam seas in Hamburg Ship Model Basin HSVA.

Visual observations and video recordings of the ship model show how the ship drifts, sways, and heaves in beam seas. Its center of gravity (COG) appears to follow a similar path to those of the water particles subject to drift and orbital motion in waves. A comparison between tests with the free-drifting and the soft-moored ship models in beam seas shows clear differences in ship survivability, obviously only because in the latter case the ship's drift motion is eliminated. Such a difference in survivability of a RoPax ship can be traced back to the accumulated water volume on the vehicle deck and finally also to the rate of net floodwater inflow onto the vehicle deck.

*3.2. Difference in Capsize Rate of a Damaged RoPax Ship in Beam Seas between Free-Drifting and Soft-Moored Condition in Model Tests*

The behavior of the MT RoPax ship with side damage was investigated in the framework of FLARE in irregular beam seas (a) in free-drifting, (b) in soft-moored conditions. In the former case, the ship was drifting abeam at speeds of ca. 0.5–0.9 m/s, depending on the significant wave height. In the latter case, the drift motion was prevented by four diagonal mooring lines, but the sway motion was allowed. Thus, the ship model kept its average lateral position in the basin. The unavoidable reducing effect of the soft-mooring on the sway motion was very small.

The lowest GM value of 3.25 m according to the current SOLAS Ch. II-1 at the studied draft of 6.1 m of the MT RoPax was used in the tests to have realistic test conditions in long-crested irregular beam seas. The free-drifting ship capsized at significant wave height $H_s$ of 7.5 m 15 times out of 20 (75%). For the soft-moored ship the capsize boundary was found to be at $H_s$ 4.5–5.0 m. In regular waves the capsize started from the wave height $H_w$ 6.0 m for the free-drifting vessel and from $H_w$ 4.5 m for the soft-moored one.

The GM value of 1.425 m was chosen for further model tests to provide a capsize boundary at more reasonable, lower wave heights in long-crested beam seas. The free-drifting ship capsized at $H_s$ 3.5 m 12 times out of 17 (70%). For the soft-moored ship the capsize boundary was found to be at $H_s$ 2.5–3.0 m. In regular waves the capsize started from $H_w$ 6.0 m for the free-drifting vessel and from $H_w$ 4.5 m for the soft-moored one, exactly as with the higher GM value [10].

Thus, in all the test cases in the model tests, there is a significant difference between the wave heights leading to capsize in free-drifting condition and in soft-moored condition. The numerical predictions for the free-drifting condition were near the experimental soft-moored values. As the main difference between free-drifting and soft-moored cases is the presence or absence of drift, the question arises whether the simple flow models in the numerical codes take the influence of the ship motions on the flow through the damage opening sufficiently into account. The question of to what extent the ship motions themselves are influenced by the drift is not investigated in this study.

*3.3. Effect of Ship Motion on Floodwater Flow through a Damage Opening during a Wave Cycle*

Figure 3 shows an example of the time histories of the signals measured in the tests shown in Figure 2. Only a small part of the time history of the FLARE benchmark test run 265 in question is shown in Figure 3, in which case the damaged ship capsized 518 s (in full scale) after the start of the test. The two dashed curves 'Wave2' and 'Wave3' show the wave elevation in front of and behind the ship, but due to the drifting of the ship model and the movement of the towing carriage, there is a deviation in the phase angle. The red curve shows the ship heave motion at the COG, which is almost identical in magnitude to the wave elevation. This curve has practically no phase lag with respect to the wave elevation at the ship centerline.

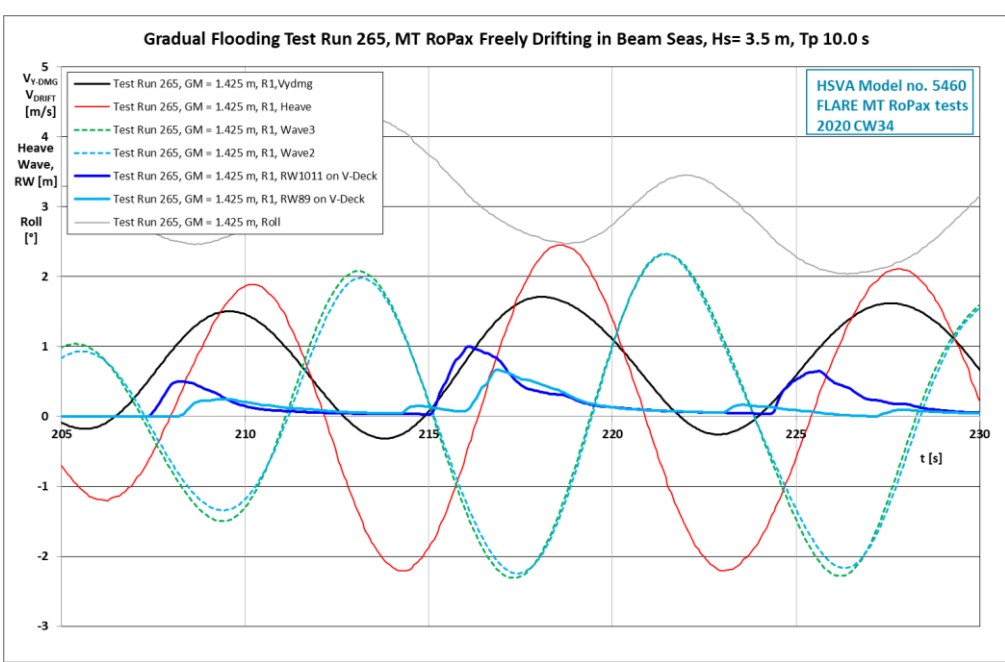

**Figure 3.** Measured time histories of the horizontal velocity of the damage opening VY-DMG, the heave motion, the relative water elevations RW1011 and RW89 on the vehicle deck at the centerline of the damage opening, and two wave elevations (Wave2, Wave3) are shown for a part of the test run 265 in beam seas.

The black curve (V$_{YDMG}$) shows the horizontal velocity of the center of the damage opening on the ship side. It mostly consists of the slowly varying ship drift, together with sway and roll motions oscillating with the wave period. The velocity is positive towards lee, and negative towards the incoming waves. The two solid blue curves show the water height on the vehicle deck at the centerline of the damage opening, which is mostly a result of the wave elevation at the damage opening, the ship heave motion and the vertical motion of the damage opening caused by the ship roll motion. The higher and darker blue curve shows the elevation near the damage opening on the ship side, whereas the lighter blue curve shows it in the middle between the ship side and the center casing. The thin gray curve above all other curves gives the ship's heeling angle, which shows values of a few degrees towards the damaged side and incoming waves. The red curve showing the heave, the two solid blue curves, the black curve showing the horizontal velocity of the damage opening, and also the gray curve have correct phase differences between them. Based on the ship motion cycles in Figure 3, we can make the following observations.

When the ship is at the wave trough (red curve), it moves slightly towards the incoming beam waves, as the negative values of the black curve show. When the incoming wave crest hits the (damaged) ship side, the ship starts to heave, the wave pressure accelerates it towards lee, the horizontal speed towards lee increases, and as the two blue curves show, water flows onto the vehicle deck. When this takes place, the horizontal speed at damage opening is towards lee and it is growing. In addition, the horizontal acceleration is positive and near its maximum towards lee side. Thus, when the floodwater flows in, the ship speeds up in the same transverse direction as the inflow. If the numerical code does not take this into account, the computed inflow can be too high, and this approximation or deficit in modeling can be a contributing factor to the in general conservative results obtained with such codes.

As the blue curves (RW1011 and RW89) in Figure 3 show, the water elevation on the vehicle deck rises suddenly as the water rushes in, but the flow speed measurements show that the flow on the deck changes direction shortly afterwards, and much of the floodwater flows out again. This repeats itself at every wave cycle, in full scale approximately every 10–20 s. Thus, the flow in the damage opening is quite unsteady.

There are two obvious points, which perhaps should be taken into account in the numerical codes when determining the flow in the damage opening.

(1) We assume the flow to be unsteady due to rapidly varying pressure heads on both sides of the opening and also due to the horizontal acceleration of the damage opening on the ship side itself. For this, the dynamic orifice equation by Lee [3] can be extended and applied.

(2) In the case of a transom stern, the speed of the ship leads to a lower water level at the transom. Thus, in analogy, for a ship drifting in beam seas, depending on the combination of sway and drift speeds, the average water level on the ship side should be slightly lower on the wave side and slightly higher on the lee side. This should have a small effect on the pressure head just outside the damage opening, which can be taken into account in modeling the inflow through the damage opening.

## 4. Steps towards Better Inflow/Outflow Boundary Condition for the Flow on the Vehicle Deck of a Damaged RoPax Ship in Waves

### 4.1. The Bernoulli Equation vs. the Dynamic Orifice Equation

Many conservative numerical predictions tend, in gradual flooding cases in beam seas, to show a slightly too rapid computed floodwater accumulation on the vehicle deck, which leads to slightly too short times to capsize. Thus, the computed floodwater net inflow through the damage opening over the wave cycles is somewhat too high.

Classically, the flow through such a damaged opening is modeled with the Bernoulli equation (BE), which describes steady flow in an inertial frame of reference. However, the ship moving in waves is not an inertial frame of reference, and the flow through the damage opening is not steady. Further, it should be noted that the flow in the opening, according to the BE, is solely dependent on the pressure difference over the opening. This means that the flow direction in the opening changes instantly together with the pressure difference without any regard for the water inertia or momentary flow speed in the opening. This is, of course, altogether not quite a realistic model for the present purpose.

As a first step to improve the inflow/outflow boundary condition, a modified version of the Dynamic Orifice Equation (DOE), as given by Lee in 2015 [3], was programmed into HSVA Rolls. In this version, also the horizontal speed and acceleration due to ship drifting and sway are taken into account in the DOE. The time-dependent flow speed or discharge (volumetric flow rate) in the opening is advanced in time together with the numerical solution of the ship and floodwater motion. The equations for all studied boundary conditions are dealt with later in Sections 4.4 and 4.5.

The DOE appears to be better suited than the BE to model the flow in the opening, but more detailed information would be needed. The problem is that the RoPax benchmark tests in FLARE show inflow values to the vehicle deck that are still lower than the results computed with BE or DOE. While the use of the DOE instead of the BE to determine the discharge in the damage opening is certainly an improvement, it appears not to be a complete solution to the inflow modeling problem on the vehicle deck.

### 4.2. The Inflow/Outflow Flow Mechanism on the Vehicle Deck as Observed in Model Tests

As the introduction of the DOE did not bring a fully satisfactory solution to the inflow modeling, the video recordings taken from the FLARE model test series were viewed anew. The cyclic inflow to and outflow from the vehicle deck is illustrated in Figures 4–6, which are individual frames in a temporal sequence of a video recorded in the HSVA tests of the damaged MT RoPax in irregular beam seas. In this context, it is worth noting that almost all side damages lead to more or less asymmetric flooding of the ship compartments. Thus, the ship is most often inclined to the damaged side.

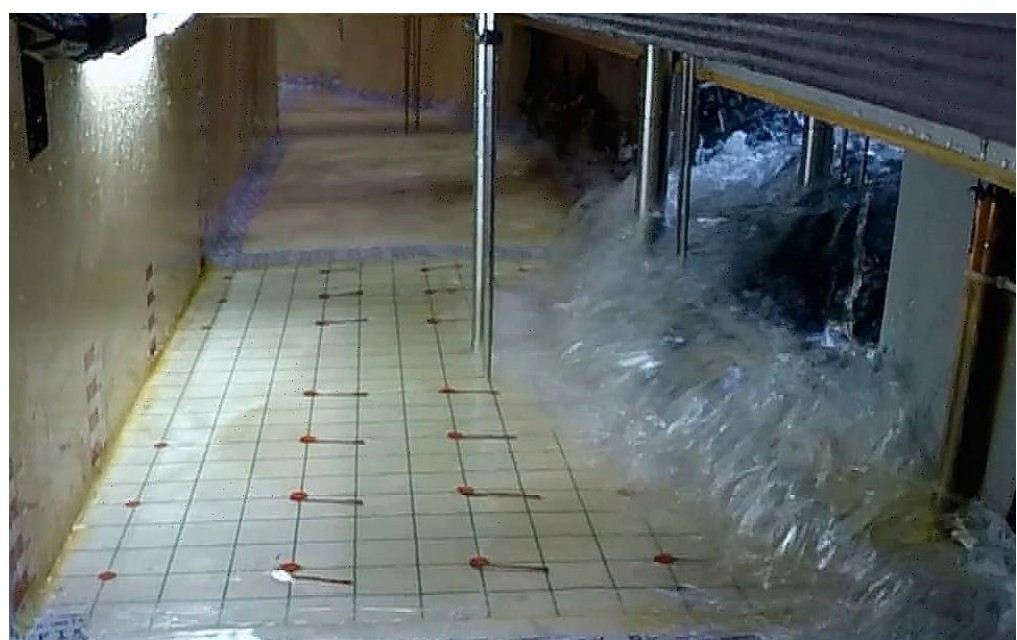

**Figure 4.** View on the vehicle deck of the ship model through a ship-fixed camera (1/3). Floodwater can be seen flowing massively in through the damage opening on the starboard, wave side of the vessel. On the deck, there are sensors for measurement of the flow speed and water elevation. The red tufts on the deck for flow direction visualization are still all pointing towards the damage opening, as a result of floodwater flown out just before the present wave came in.

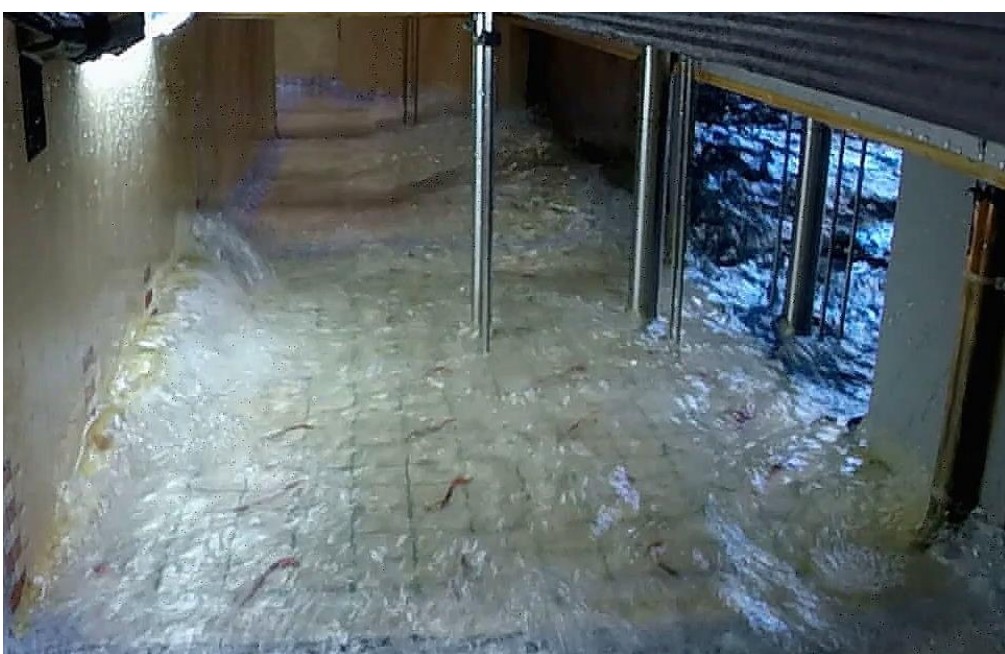

**Figure 5.** View on the vehicle deck of the ship model (2/3). Floodwater has reached its momentary maximum extent, and it starts to flow back towards the damage opening. The red tufts for flow visualization are still pointing towards the center casing and towards ship fore and aft directions, as a result of the floodwater that just flowed in.

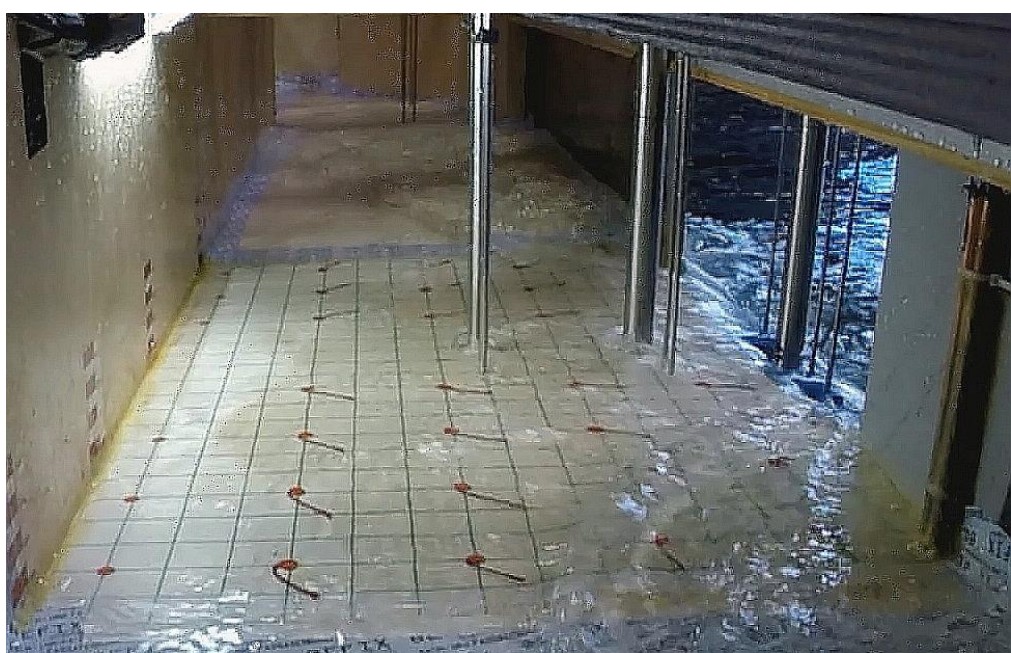

**Figure 6.** View on the vehicle deck of the ship model (3/3). Floodwater is flowing back out of the damaged opening. The red tufts for flow visualization are pointing again towards the damage opening, as a result of the floodwater flowing out. As the view is given by a ship-fixed camera, the ship heeling angle is not really visible, but can be inferred from the floodwater accumulating towards the starboard side of the deck.

Typically, the video recordings of the FLARE ship damage stability model tests show the following:

(1) ***In regular beam waves,*** the wave crests hit the ship side and water flows through the damage opening onto the vehicle deck of the RoPax ship and, of course, also into the damaged compartments below. Once the crest has passed the damaged ship side, the water on the vehicle deck flows back along the downwardly inclined deck towards the damage opening and further through the opening out of the vehicle deck. With the next regular wave, this process is repeated anew.

(2) ***In irregular beam seas,*** the highest wave crests bring water onto the vehicle deck and into the damaged compartments below. In between these high wave crests there are lower wave crests and wave troughs, which do not bring any water onto the vehicle deck, as the water elevation at the damage opening does not reach the vehicle deck level at the opening. During these relatively long periods between higher wave crests, the floodwater mostly flows along the inclined vehicle deck out of the damage opening back to the sea.

*4.3. Outflow through the Damage Opening on the Heeled Vehicle Deck*

When the water level just outside of the damage opening lies below the level of the vehicle deck, the Bernoulli equation gives the floodwater outflow speed at the opening solely based on the height of the fairly thin floodwater layer on the vehicle deck at the damage opening, as shown on the left-hand side (LHS) in Figure 7. However, in a damaged ship, the flow speed of the whole water layer developing along the usually inclined deck can due to gravity, lead to a higher floodwater discharge towards the damage opening than the outflow discharge through it described by the BE. See the case on the right-hand-side (RHS) in Figure 7.

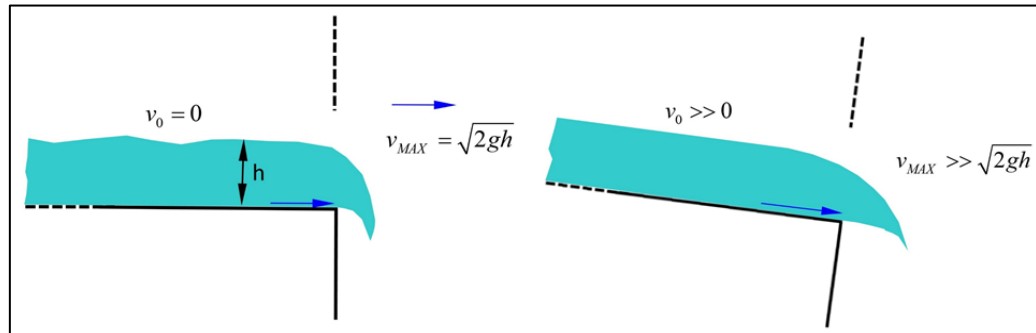

**Figure 7.** The flow on the horizontal and inclined vehicle decks at the damage opening.

In the theoretical and numerical modeling, this can lead to a non-realistic accumulation of floodwater in front of the damage opening, as the floodwater flows out according to BE at a smaller rate than what flows onto this spot from elsewhere along the inclined deck. Thus in the numerical model, when the floodwater does not flow out at a sufficient rate, the water level rises locally and the floodwater flows on the deck and in the ship's longitudinal direction to both sides of the damage opening, towards the bow and stern of the vessel. Altogether, this leads to a reduced outflow of the floodwater and to a somewhat too rapid floodwater accumulation on the vehicle deck. The consequence is also a somewhat too short numerically computed Time to Capsize (TTC).

It is difficult to make reliable measurements on the overall flow speeds on the deck of a damaged RoPax ship model in beam seas that would allow meaningful conclusions. For this reason, a separate 2D test rig was used to measure the effect of the deck inclination on the outflow speed. These tests are described in Appendix A. The main conclusion based on the results with the test rig is that the deck inclination has a significant effect on the speed of the shallow water flow on the deck and thus also on the floodwater outflow speed through the damage opening. For details, see Appendix A.

The observations on the damaged ship model in the seakeeping tests and the measurements with the small test rig (see Appendix A) call in the numerical modeling for taking the flow speed of the floodwater layer on the inclined deck into account also in the boundary condition for the damage opening to an open deck.

### 4.4. Formulations for Floodwater Discharge at the Damage Opening

For purposes of numerical testing, the DOE was programmed into HSVA Rolls, which enables comparison with the results computed using the BE for modeling the water inflow/outflow through the damage opening.

The BE is an equation for steady flow that neither takes the water inertia in the opening into account nor does it regard the effect of the prevailing flow speed in the opening. As the flow direction in the opening, at least in model tests, continuously changes when the ship floats in waves, it is expected that the Bernoulli equation reacts too rapidly to the changes in the water level on both sides of the opening. This tends to lead to too high flow rates in the opening. The classical Bernoulli equation to model the flow in the damage opening can in the simplest case be written as:

$$\frac{1}{2}u^2 = \frac{(p_0 - p_1)}{\rho} \quad , \tag{1}$$

in which $p_0$ and $p_1$ are the pressure values on different sides of the damage opening, $\rho$ the fluid density and $u$ the flow velocity in the opening. For the flow velocity we get

$$u = \sqrt{\frac{2(p_0 - p_1)}{\rho}} \quad , \tag{2}$$

and for the discharge Q through the damage opening

$$Q = \mu A \sqrt{\frac{2(p_0 - p_1)}{\rho}} \quad , \tag{3}$$

where $A$ is the area of the opening and $\mu$ the discharge coefficient. Instead of using only the classical Bernoulli equation to model the flow in the damage opening, also the following dynamic orifice equation by Lee [3] is considered:

$$\frac{\sqrt{A}}{2}\frac{\partial u}{\partial t} + \frac{7}{8}u^2 = \frac{(p_0 - p_1)}{\rho} \quad . \tag{4}$$

The DOE was programmed into HSVA Rolls, using a formulation based on discharge $Q$ (=$\mu A u$), instead of just the flow velocity $u$ in the opening. The following formulation shows the simplest case of the classical formulations for Q through a damaged opening:

$$Q_{NEW} = \underbrace{\mu A \bar{u}_{OLD}}_{Q_{OLD}} + \Delta t \mu \left( \frac{\sqrt{A}}{\rho}[2\Delta p_a + \Delta p_{12}] - \frac{7\sqrt{A}}{4}\bar{u}_{OLD}^2 - \sqrt{A}\,v_{YD}^2 - A a_{YD} \right), \tag{5}$$

in which $\Delta p_a$ and $\Delta p_{12}$ describe in HSVA Rolls the vertically varying pressure differences over the opening. The two additional terms at the end of the equation are related to the horizontal velocity $v_{YD}$ and the horizontal acceleration $a_{YD}$ of the damage opening. Due to the transverse horizontal velocity $v_{YD}$ consisting of drift and sway speeds of the ship mostly in beam seas, the water pressure and thus the water level outside the damaged ship side can slightly change. This is in analogy with the water level lowering at the transom stern of an advancing ship.

The horizontal acceleration $a_{YD}$ at the damage opening due to the ship motion is connected to an added mass term, and together these have an effect on the development of the flow at the opening. As can clearly be seen in eq. (5) used in the numerical test runs, the discharge $Q$ is modified at every time step by the term in brackets on the RHS, but is not solely determined by it. Thus, any change in the flow direction is better modeled with the DOE than with the BE, which determines the flow at the damage opening purely based on the pressure difference on the opposite sides of the opening. With the DOE, the inflow/outflow in the damage opening are less abrupt, and the water height just inside the opening changes more gradually. The net flow rate into the ship through the damage opening was in the studied cases in beam seas clearly lower than the one obtained using the BE. Thus, the time to capsize is somewhat longer.

The use of the DOE as applied in (5) appears to be better suited than the use of the BE to model the flow in the damage opening. However, in spite of this small progress, the RoPax benchmark tests in FLARE appear to show net floodwater inflow values to the vehicle deck, which are still lower than the results computed with BE or DOE.

### 4.5. Improved Boundary Condition for the Damage Opening to an Open Deck

Leaning on the observations made in the model tests and on the first numerical results obtained with the DOE, an improved inflow-outflow model for the damage opening on the vehicle deck was developed. This formulation takes the speed of the water flow on the inclined deck at the damage opening into account, based on the use of the numerical solution of the shallow-water-equations SWE on the deck: Let us consider inflow to and outflow from the vehicle deck in the following two cases.

(1) ***The roll or heeling angle is negative or zero***. The floodwater flows from the damage opening on the starboard side towards the center casing.
    The flow speed at the damage opening is determined with Bernoulli Equation or with Dynamic Orifice Equation, and the change in the linear momentum due to the water

inflow in the opening is taken into account as a boundary condition in the numerical solution of the SWE on the vehicle deck. As water outside the damage opening can be assumed to have practically zero speed, this formulation is a quite proper and suitable approximation. This, along with BE, is the original model in HSVA Rolls.

(2) *The roll or heeling angle is positive*. The floodwater flows from the inner parts of the vehicle deck downward along the inclined deck to the damage opening and further out to the sea below.

In this case the floodwater on the deck as a shallow-water layer can develop a significant speed towards the damage opening. This can be seen in model tests with a RoPax ship with side damage in beam seas. The measured flow speed data obtained with a test rig can be found in Appendix A. In numerical simulations the flow speed on the inclined deck can be determined with SWE. The speed at the damage opening can be taken as a combination of the speed given by the SWE and that given by the BE or DOE based on the water level difference in the opening.

With this boundary condition in the numerical solution, the floodwater flowing down along the inclined vehicle deck can be better taken into account than solely with BE or DOE. The modeling should be particularly important in irregular seas, in which there are long periods of floodwater outflow between the occasional higher wave crests that bring water onto the vehicle deck.

The improved boundary condition defines the speed at the damage opening as a combination of the speed given by the SWE and that given by the BE or DOE. Incorporating the floodwater flow speed on the inclined deck into the Bernoulli-based boundary condition at the damage opening is certainly possible, but somewhat cumbersome in the program structure of HSVA Rolls. For this reason, the floodwater flowing speed on the inclined deck was combined with the DOE, which altogether is a more refined model. This improved boundary condition is here called SWEDOE. The following formulation was used in the test simulations:

$$Q_{NEW} = \underbrace{\mu A \bar{u}_{OLD}}_{Q_{OLD}} + \Delta t \mu \left( \frac{\sqrt{A}}{\rho} \left[ 2\Delta p_a + \Delta p_{12} \right] - \frac{7\sqrt{A}}{4} \bar{u}_{OLD}^2 - \sqrt{A} \, v_{YD}^2 \atop + \sqrt{A} \, v_{SWE}^2 - A a_{YD} \right) . \tag{6}$$

The new additional term on the RHS of the equation is related to the horizontal velocity $v_{SWE}$ on the vehicle deck just inside of the damage opening. Thus, for modeling the inflow/outflow in Cases 1 and 2, the Equations (5) and (6), respectively, are applied.

Taking the sign definitions in HSVA Rolls into account, the specific formulation in HSVA Rolls is acquired:

$$Q_{NEW} = \mu A \bar{u}_{OLD} + \Delta t \mu \left( \sqrt{A} \left[ \frac{1}{\rho} \left[ 2\Delta p_a + \Delta p_{12} \right] - \frac{7}{4} |\bar{u}_{OLD}| \bar{u}_{OLD} \atop - |v_{YD}| v_{YD} - |v_{SWE}| v_{SWE} \right] - A a_{YD} \right) . \tag{7}$$

## 5. Comparison of the Computed Results with FLARE Benchmark Test Experimental Data

The three different formulations for boundary conditions BE, DOE, and SWEDOE were tested in the program HSVA Rolls, first with the MT RoPax in regular beam waves to study the time histories of the heeling angle until capsize. Second, this was repeated in irregular beam seas to study the effect of the chosen boundary condition on the ship survivability and the computed Time to Capsize TTC. The damage case is the FLARE RoPax benchmark damage case, illustrated in Figure 2. The damaged compartments below the vehicle deck extend over the whole breadth of the vessel; see [1] for details, which makes the case rather difficult to compute. In all numerical simulations the sway is computed,

but the drift velocity has a value measured in the experiments. This value depends mainly on the wave height. Figure 8 shows the computed curves for the three boundary conditions and the corresponding two experimental curves. The following observations are based on a few similar comparisons as the one in Figure 8:

In all cases the computed water ingress onto the vehicle deck is in general more linear in time and generally larger than in the experiments. As a consequence, the roll angle grows more evenly than in the experiments.

- The use of the BE as a boundary condition is easy, and the ship heeling process is in this case quite linear. Using the classical value for the discharge coefficient (0.6) leads to too short a time to capsize. With reduction of the value of the discharge coefficient, the TTC could be prolonged. This may lead to practical results, but it does not reflect the prevailing physics too well.

- The use of the DOE as a boundary condition requires time-integration of the flood-water discharge through the damage opening simultaneously with the time-integration of the ship motions in the simulation program. The programmed boundary condition tested gives the discharge in the opening also as a function of the ship horizontal and transverse acceleration and speed at the damage opening. Thus, the lateral ship motions influence the flow in the opening, which appears to lead to a less smooth development of the ship roll angle than in the case of the BE. In transient cases the DOE is a much more physically correct boundary condition than the BE. The use of the DOE delays capsizing in comparison with the BE. However, as several ship motion components influence the flow in the damage opening, the flow can also be more easily distorted if the ship motions are not accurately predicted.

- The use of the SWEDOE requires time-integration of the discharge through the damage opening and the input of the flow speed on the vehicle deck into the boundary condition. Also in this formulation, the lateral ship motions are taken into account. In most cases the inclusion of the outflow speed on the inclined deck in the boundary condition reduced the net inflow onto the vehicle deck, delayed capsize, and had a prolonging effect on the time to capsize TTC. Thus, the SWEDOE curve showing the development of the roll angle over time is similar to the DOE curve but more gradual. In some cases, the SWEDOE formulation postpones capsize considerably, when large amounts of floodwater flow out of the vehicle deck.

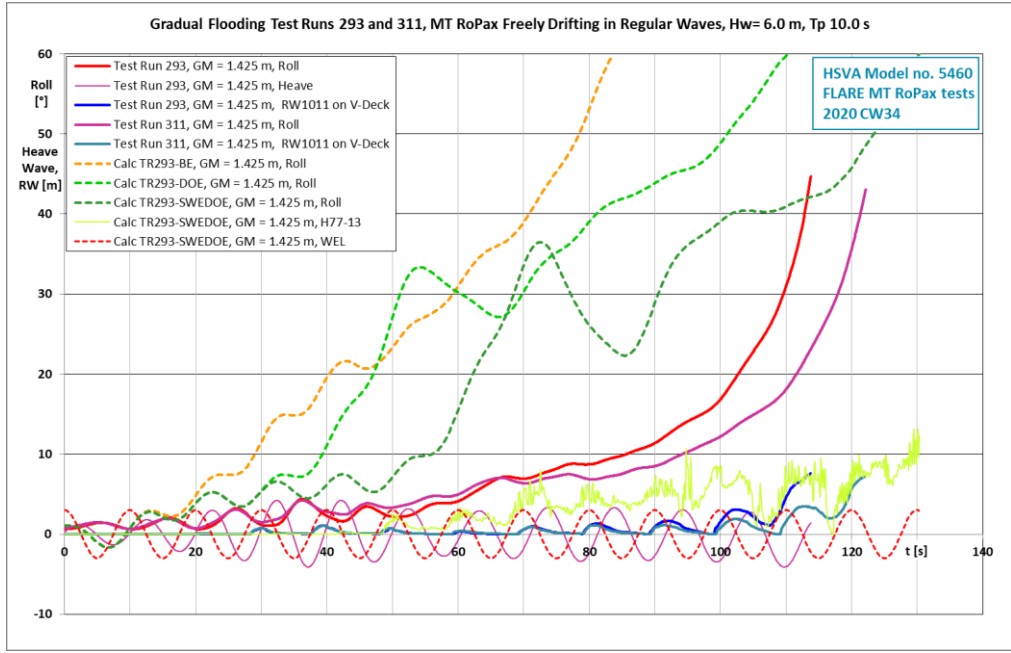

**Figure 8.** Computed and experimental time histories of the roll angle of the ship in regular beam waves with 6.0 m wave height, shown until capsize. The dashed curves at the center show the computed results obtained using the boundary conditions BE, DOE, and SWEDOE.

The three boundary conditions in the numerical model were also tested in irregular beam seas at two wave heights and compared with experimental values. The results are shown in Tables 2 and 3. The data shows average values of 20 computations as well as of 20 experiments.

**Table 2.** Computed Times to Capsize TTC in percent in comparison with experimental values.

| FLARE MT RoPax Benchmark Damage Case | | | | | |
|---|---|---|---|---|---|
| GM | $H_s$ | BE | DOE | SWEDOE | EXP |
| 1.425 m | 3.5 m | 28.3% | 53.1% | 63.1% | 100% |
| 3.250 m | 7.5 m | 36.7% | 96.3% | 96.6% | 100% |

**Table 3.** Computed survival rates in comparison with experimental values.

| FLARE MT RoPax Benchmark Damage Case | | | | | |
|---|---|---|---|---|---|
| GM | $H_s$ | BE | DOE | SWEDOE | EXP |
| 1.425 m | 3.5 m | 0/20 | 3/20 | 4/20 | 7/20 |
| 3.250 m | 7.5 m | 0/20 | 17/20 | 18/20 | 5/20 |

Based on the data in Tables 2 and 3, the following conclusions can be drawn.

- The boundary conditions at DOE and SWEDOE yield better values for the Time to Capsize TTC than the BE.
- In the lower sea state with $H_s$ 3.5 m, the boundary conditions of DOE and SWEDOE yield better values for the survival rate than the BE. The survival rate given by BE is too low.
- In the higher sea state with $H_s$ 7.5 m, the boundary conditions of DOE and SWEDOE yield too high survival rates. In addition, in this case, the BE gives a too low survival rate.
- Thus, the use of the DOE and SWEDOE are certainly steps in the direction of a better boundary condition for the damage openings, but the formulations used in this brief study do not yet lead to very satisfactory results.

## 6. Comparison of the Computed Results with FLARE Flood Mitigation Test Experimental Data

The three formulations for boundary conditions BE, DOE, and SWEDOE were further tested in the program HSVA Rolls using the MT RoPax model in irregular beam seas to study the effect of the chosen boundary condition on the ship's survivability and the computed Time to Capsize TTC. This second damage case is the FLARE MSRC[1] Damage Case 2 on the MT RoPax, originally used for testing active flooding mitigation with counter flooding see [8,11] for details. It is limited only to the starboard side of the ship, which makes it easier to compute than the previous benchmark test case. Experimental data obtained without any flooding mitigation was used for comparison. The three boundary conditions in the numerical model were tested in irregular beam seas at two wave heights. The results are shown in Tables 4 and 5. The data shows average values of 20 computations and 10 experiments.

**Table 4.** Computed Times to Capsize TTC in percent in comparison with experimental values.

| FLARE MT RoPax MSRC Damage Case 2 for Flooding Mitigation | | | | | |
|---|---|---|---|---|---|
| GM | $H_S$ | BE | DOE | SWEDOE | EXP |
| 3.40 m | 3.5 m | 38.0% | 77.7% | 93.2% | 100% |
| 3.40 m | 5.0 m | 54.4% | 87.4% | 82.5% | 100% |

**Table 5.** Computed Survival rates in comparison with experimental values.

| FLARE MT RoPax MSRC Damage Case 2 for Flooding Mitigation | | | | | |
|---|---|---|---|---|---|
| GM | $H_S$ | BE | DOE | SWEDOE | EXP |
| 3.40 m | 3.5 m | 0/20 | 2/20 | 3/20 | 4/20 |
| 3.40 m | 5.0 m | 0/20 | 0/20 | 0/20 | 0/20 |

Based on the data in Tables 4 and 5, the following conclusions can be drawn.

- The boundary conditions at DOE and SWEDOE yield much better values (77.7–93.2%) for the Time to Capsize TTC than the BE (38.0–54.4%).
- In the lower sea state with $H_S$ 3.5 m, the boundary conditions of DOE and SWEDOE yield better values for the survival rate than the BE. The survival rate given by BE is too low.
- In the higher sea state with $H_S$ 5.0 m, all boundary conditions BE, DOE, and SWEDOE yield zero survival rates, like the experiments do.
- The use of the DOE and SWEDOE are certainly steps in the direction of a better boundary condition for the damage opening to the vehicle deck. In this damage case the results simulated using DOE or SWEDOE are already quite satisfactory.

Figures 9 and 10 give an impression of the time-histories of the heeling angle computed using the boundary condition SWEDOE, together with time-histories measured in the HSVA experiments. The wave trains used in the computations and the model tests are not identical. The following observations can be made:

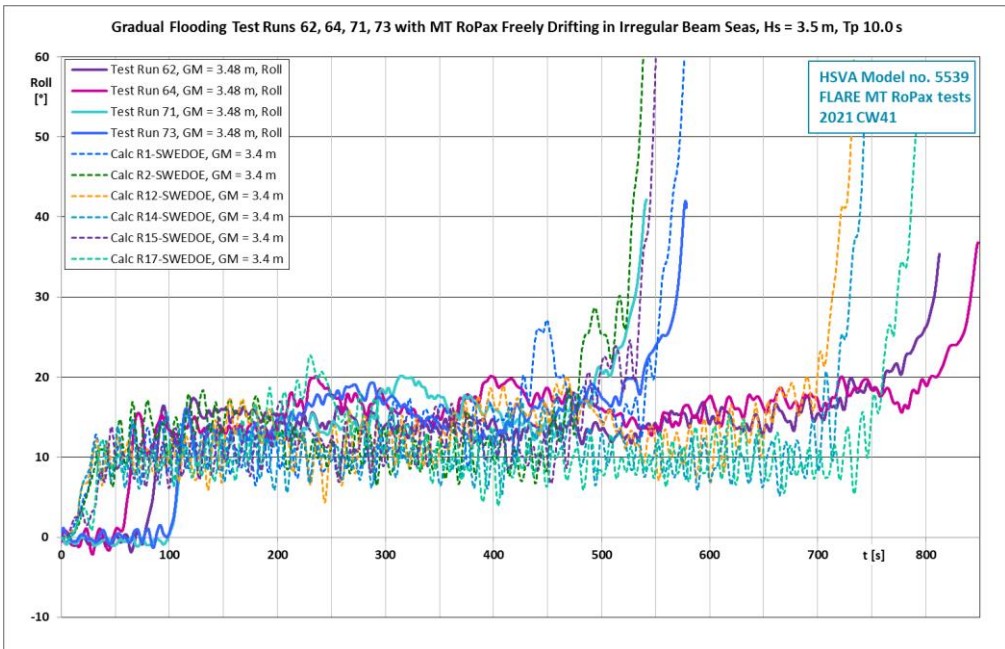

**Figure 9.** Computed and experimental time histories of the roll angle of the ship in irregular beam seas with $H_S$ 3.5 m and $T_P$ 10.0 s, shown until capsize. The computed curves show the results obtained using the boundary condition SWEDOE.

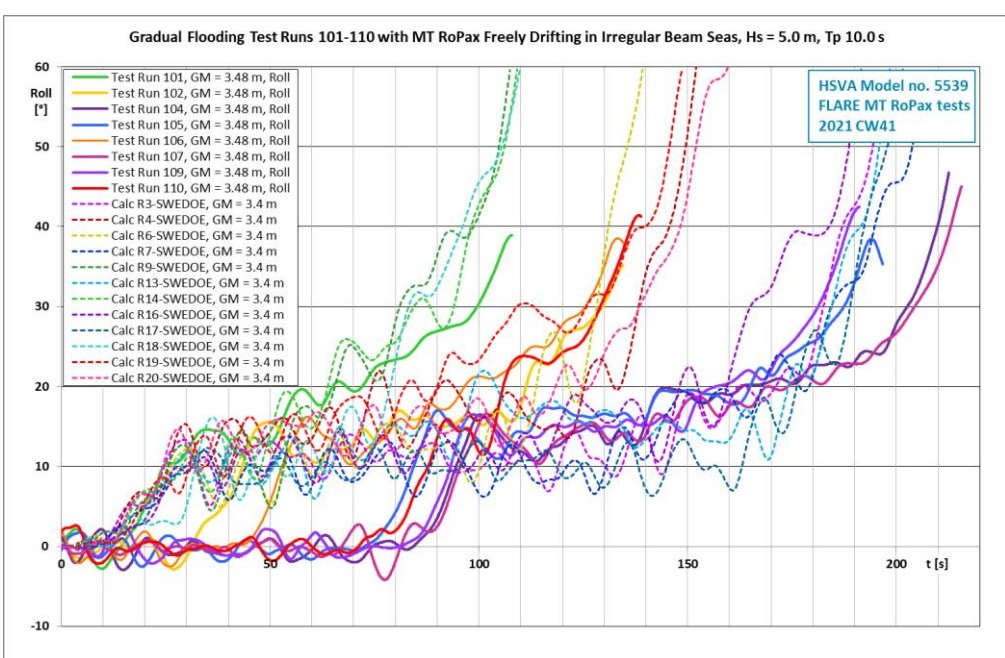

**Figure 10.** Computed and experimental time histories of the roll angle of the ship in irregular beam seas with $H_S$ 5.0 m and $T_P$ 10.0 s, shown until capsize. The computed curves show the results obtained using the boundary condition SWEDOE.

In the experiments, the heeling angle rises from around zero to another level later than in the computations. This is likely to be related to the physical wave train realization in the towing tank, in which the development to full wave height can be slightly delayed at the beginning of the wave train.

The oscillations in the heeling angle due to wave action and the sloshing of floodwater in the internal compartments are more pronounced in the (inviscid) numerical computations than in the experiments.

Altogether, the comparison of the computed results with the experimental curves yields more favorable results in the mitigation test case shown in Figures 9 and 10 than in the benchmark test case shown in Figure 6. This is related to the damaged compartments below the vehicle deck extending over the whole breadth of the ship in the benchmark damage case, which can occasionally lead to heavy sloshing in these compartments. In the mitigation test case, the damage is limited to the starboard side only, which makes it easier to simulate the case.

## 7. Discussion

It is a challenging task to numerically model the behavior of a damaged ship floating in waves together with the associated flooding process. While modeling with Reynolds Averaged Navier Stoke Equations (RANSE) has so far not yielded many practical results, hydraulic or quasi-hydrostatic flooding models together with hydrostatic ship stability and strip-theory based calculations yield results, but not always with the desired reliability and accuracy. The more ship and floodwater dynamics through wave action are involved, the more unreliable the results obtained with these models tend to become. For cases of gradual progressive flooding, typical of a cruise ship with many small compartments, such models can yield satisfactory results. For ships with large damaged open spaces, floodwater and ship dynamics are more important, and consequently, more sophisticated modeling of ship and floodwater dynamics is necessary. The physical effects included (✓) in the different formulations studied are summarized in Table 6.

**Table 6.** Physical effects included in different boundary condition models.

| Boundary Condition for the Damage Opening on Ship Side | Type | | |
|---|---|---|---|
| **Effect Modeled** | **BE** | **DOE** | **SWEDOE** |
| Pressure or water height difference | ✓ | ✓ | ✓ |
| Flow speed in the opening | | ✓ | ✓ |
| Horizontal drift + sway velocity of the ship | | ✓ | ✓ |
| Horizontal acceleration of the damage opening | | ✓ | ✓ |
| Floodwater inertia in the opening | | ✓ | ✓ |
| Shallow water speed on deck in front of the opening | | | ✓ |
| Simple formulation with no memory effect | ✓ | | |
| Time-integration of the flow speed in the opening | | ✓ | ✓ |
| Water viscosity on deck or in the damage opening | | | |

In many programs, the ship motions are at least in part based on modeling with strip theory or with a panel method, both resting on assumptions of potential theory. Application of these requires the use of additional empirical roll damping coefficients, which depend also on the ship floating position, particularly on the heeling angle [12]. When there is floodwater on the ship, the floodwater motion can dampen or excite the ship roll motions, exactly as an anti-roll tank in a ship does. Further, whether the floodwater sloshes in a closed compartment or in a compartment open to the sea through the damage opening, can have a significant effect on the roll damping of the ship-floodwater system [11].

Empirical roll damping coefficients for the whole ship are not really sufficient to model the additional roll damping or excitation caused by the floodwater. Therefore, it is important to try to model the floodwater behavior on the ship as well as possible. For this, a numerical model of the dynamic behavior of the floodwater is needed, together with a suitable boundary condition for the damage opening, which describes the inflow/outflow not only in the steady flow case.

HSVA Rolls uses shallow-water-equations (SWE) to model the dynamic floodwater flow on decks. In order to improve the accuracy of the simulations, the effect of different boundary conditions on the floodwater discharge through the damage opening was studied. There is a clear need to take the dynamic character of the flow in the damage opening better into account than what the classical BE does.

## 8. Conclusions

Tests with scale models of damaged ships floating in waves show the unsteady character of the flow in the damage openings and in the interior compartments. The obvious difference between this and the very simple, mostly hydraulic modelling of the floodwater in the ship interior, used in most numerical models for damaged ship survivability, raises the question of the suitability of such simple models such as the Bernoulli equation for the flow in the damage opening.

In FLARE model tests, a significant difference was found between the times to capsize in beam seas in free-drifting cases and in soft-moored cases. As the main difference between these two cases is the presence or absence of ship drift, it was assumed important to try to take the ship motions into account in modeling the flow through the damage opening. The phase difference between the floodwater inflow and the lateral motion at the damage opening appears to be small in beam seas: When the floodwater flows in, the ship speeds up in the same transverse direction as the inflow, which has a reducing effect on the inflow.

In order to provide an alternative for the BE, the DOE used by Lee [3] in 2015 was modified and extended for the horizontal motion at the damage opening on the ship side. This equation takes the inertia of the floodwater in the damage opening into account, and its use led in this investigation to a reduced net water ingress in the ship in comparison with the BE and to a somewhat longer time for the ship to capsize or to flood. With this

desired effect, the use of a boundary condition for the flow at the damage opening that takes the fluid inertia and the motion of the opening into account is an improvement in comparison with the use of the classical BE. The additional computational effort for the use of the boundary conditions DOE or SWEDOE instead of BE was insignificant.

In heeled internal compartments and particularly on vehicle decks, the flow of a relatively thin layer of floodwater on an inclined floor or deck can due to the gravity component develop very significant speeds. This is a typical case for the outflow through a damaged opening on the vehicle deck of a RoPax ship. Proper modeling of this requires the determination of the speed of the floodwater on the inclined deck with a suitable numerical method and the input of this speed in the boundary condition at the damage opening. As presently applied in HSVA Rolls, this led in the simulated cases to a slightly lower net rate of water ingress to the vehicle deck and thus to slightly longer survival times in beam seas.

Based on the achieved results, further study and application of boundary conditions for the damage opening, which take the floodwater inertia and the ship motions better into account than the Bernoulli equation does, are recommended for survivability simulations of damaged ships in waves.

**Funding:** The research presented in this paper was carried out in the framework of the project Flooding Accident Response (FLARE) no. 814753, under the H2020 program funded by the European Union, which is gratefully acknowledged. All opinions are solely those of the author.

**Institutional Review Board Statement:** Not applicable.

**Informed Consent Statement:** Not applicable.

**Data Availability Statement:** Not applicable.

**Acknowledgments:** The research presented would have hardly been possible without the framework of the EU-project FLARE, and it has certainly benefited from the open scientific communication between various distinguished colleagues in this framework and outside of it.

**Conflicts of Interest:** The author declares no conflict of interest.

## Appendix A. Model Tests for Floodwater Outflow

In order to shed light on the validity of the Bernoulli model for the inflow and outflow through the damage opening on a RoPax ship in waves, model tests with a small test rig were carried out in HSVA in the framework of the EU-project Flooding Accident Response (FLARE). The test rig consists of (1) a water tank on the left, (2) a low rectangular opening and gate of full width for the water flow out of the tank to a deck, and (3) the said deck on the right. See Figures A1 and A2. On the deck the development of the water height and of the flow speed were measured at four sections $S_1$–$S_4$. The rig was heeled to 0°, 5°, 10°, 15°, and 20° inclination angles to provide the deck with a slope, as in a heeling ship. Three different initial water level heights in the tank were used in the tests.

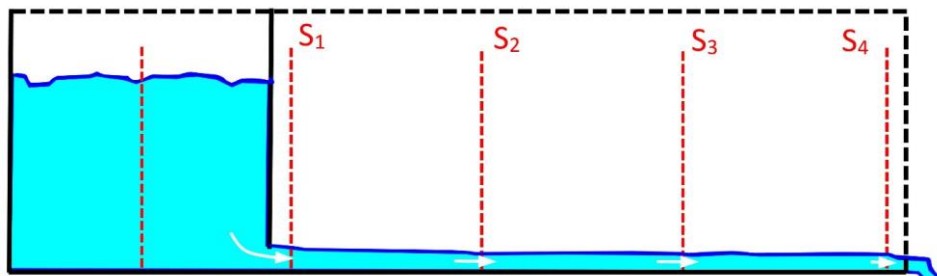

**Figure A1.** Test rig for outflow tests. The instrumented four sections are denoted $S_1$...$S_4$, from left to right.

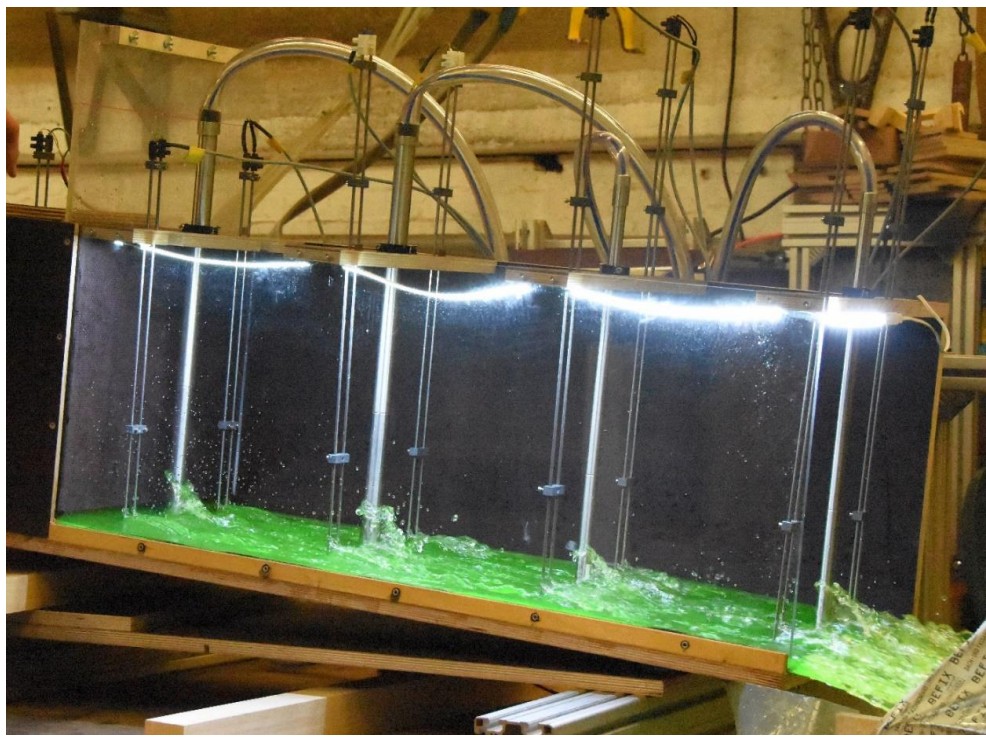

**Figure A2.** Outflow test with an inclination of 5° and initial water depth of 0.15 m in the tank.

The purpose was to investigate how well the BE describes the time-dependent flow in the test cases with different initial water heights and deck inclination values, when the gate between the tank and the deck is suddenly opened. This simple test arrangement should give insight on the suitability of the BE to describe the floodwater inflow and outflow through a damage opening onto the vehicle deck of a damaged RoPax ship in waves.

The model test results were scaled up to full scale with a ratio of 1:14. With this scale, the width of the deck on the test rig corresponds to the width between the ship side and the center casing of FLARE Ship No. 6, i.e., the MT RoPax, and amounts to 13.72 m. The initial water depths in the tank amount to the extreme 4.2 m, the moderate 2.1 m, and the fairly low 0.7 m, providing a wide range of pressure heads, well covering the expected range of these values in low to moderate sea states for the typical RoPax in question, also when significant transverse floodwater sloshing takes place on the deck.

The measured flow speed in section $S_1$ is shown in Figure A3 with the solid red curve. The solid blue curve shows the water level in the tank. The two dashed curves RWS1 and RWS2 show the water level at section $S_1$. The dashed curve V1-BE shows the flow speed at the section $S_1$ according to the Bernoulli model based on the difference in the water level in the tank and the average given by the two water level sensors in the section $S_1$.

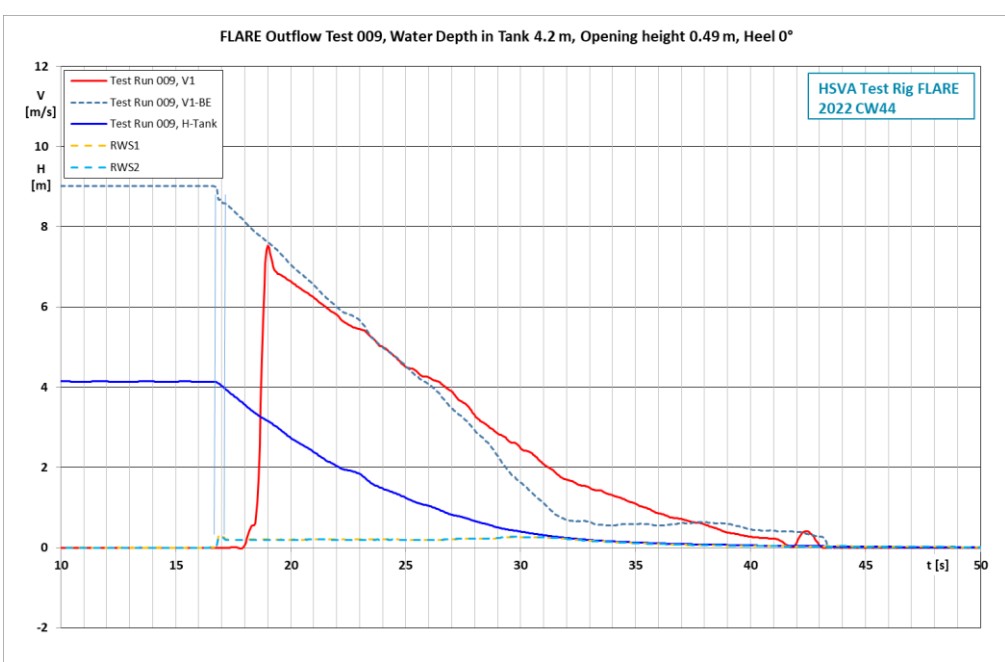

**Figure A3.** Flow speed $V_1$ and water elevations RWS1 and RSW2 at the first section $S_1$ for the test run 009.

As shown in Figure A3, the relative wave sensors RWS1 and RWS2 on both sides of the flow sensor $V_1$ show the rise of the water front in section $S_1$ at ca. 1.0–1.2 s (in f.sc.) before the flow sensor shows the rise in the flow velocity. This delay is assumed to be the reaction time of the flow sensor, and it is visible in all measurements. The fluid starts at rest when the gate is suddenly opened, and after the initial delay in the flow sensor the flow reaches its measured full speed rapidly, but not instantly. Depending on the case, this rise-time to full flow speed amounts to ca. 1.0–1.3 s, which is something the BE completely ignores.

Such a rise-time of the flow speed on a damage opening on the ship side should be assessed in relation to the ship roll period or the modal period of the relevant sea state, in this case around 10 s, during which the flow in the damage opening changes its direction twice. That is, once every 5 s. The delay of 1.0–1.3 s in the rise-time during the period of ca. 5 s can be considered significant for good accuracy in modeling the flow in the damage opening.

Another feature of the BE is also problematic: when the pressure head, or water height difference, is zero, the flow speed according to the BE is also zero. In the experiments this is of course not the case. The flow does not start instantly, but from rest with a small delay, when the gate is opened. The fluid motion gains momentum, and the fluid inertia mostly keeps its momentum and speed, also after the pressure head diminishes. Thus due to a lack of inertia, the flow speed predicted by the BE has a phase difference with respect to the measured values. When the deck is horizontal, mainly viscous effects have a reducing effect on the fluid momentum and thus also on the flow speed. The somewhat premature end of flow according to the BE, is clearly visible in the diagrams.

Beside the measured values in the fourth section $S_4$ as in the preceding figure, the additional dashed red curve (velocity $V_4$ computed as BC) in Figure A4 shows also the outflow velocity computed with the typical boundary condition for the damage opening based on BE. That is, based only on the water elevation difference inside (at $S_4$) and outside of the opening for outflow at right. The water elevation outside is zero, when the free surface lies below the deck level. The measured flow speed value ($V_4$, shown by the violet curve) at $S_4$ lies initially higher than the red computed value but decreases faster. The velocity $V_4$ computed as a boundary condition shows higher values as long as there is any

water on the deck, because the water elevation outside is zero. Thus, the classical boundary condition based on BE describing the (inviscid) flow through the damage opening (a) does not in general take the flow speed on the vehicle deck into account, and (b) may show too high flow values, when there is a thin layer of water on a horizontal deck.

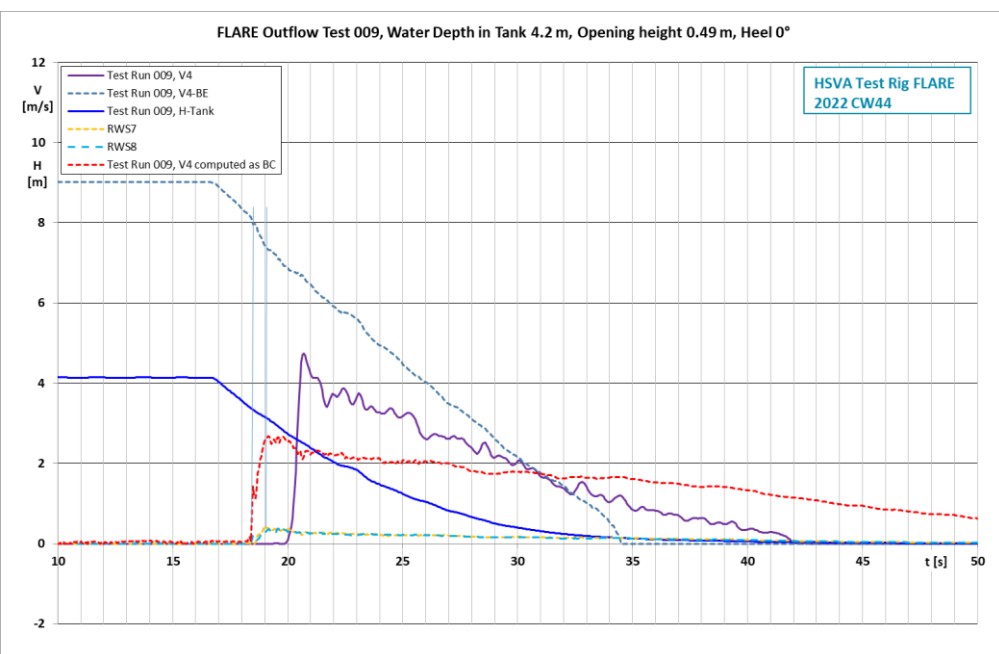

**Figure A4.** Flow speed $V_4$ and water elevations RWS7 and RSW8 at the fourth section $S_4$ for the test run 009. In addition, the red dashed curve shows the flow speed $V_4$ computed with BE as the typical boundary condition at the damage opening.

Figure A5 shows also the water discharge volume (Vol4), (1) based on the flow speed $V_4$ computed as a boundary condition with BE, and (2) based on the measured flow speed $V_4$. These two curves coincide only in this particular case. If the initial water depth in the tank is lower than in this test case 009, the viscous reduction in the flow speed is more pronounced, and the BE overpredicts the discharge.

If the heeling angle has even a small positive value, the measured discharge volume out of the deck is much higher than the value predicted by the boundary condition using BE, which is based only on the water height difference over the damage opening. Such a test case with an inclination angle of 15° is shown in Figure A6. These two curves show widely different values due to the heeling angle, which speeds up the flow, but which is not taken into account in the BE. The flow speeds calculated with BE and the measured flow speeds and discharge values show now very different values, which indicates the need to take the speed of the shallow water flow on decks into account in the boundary condition.

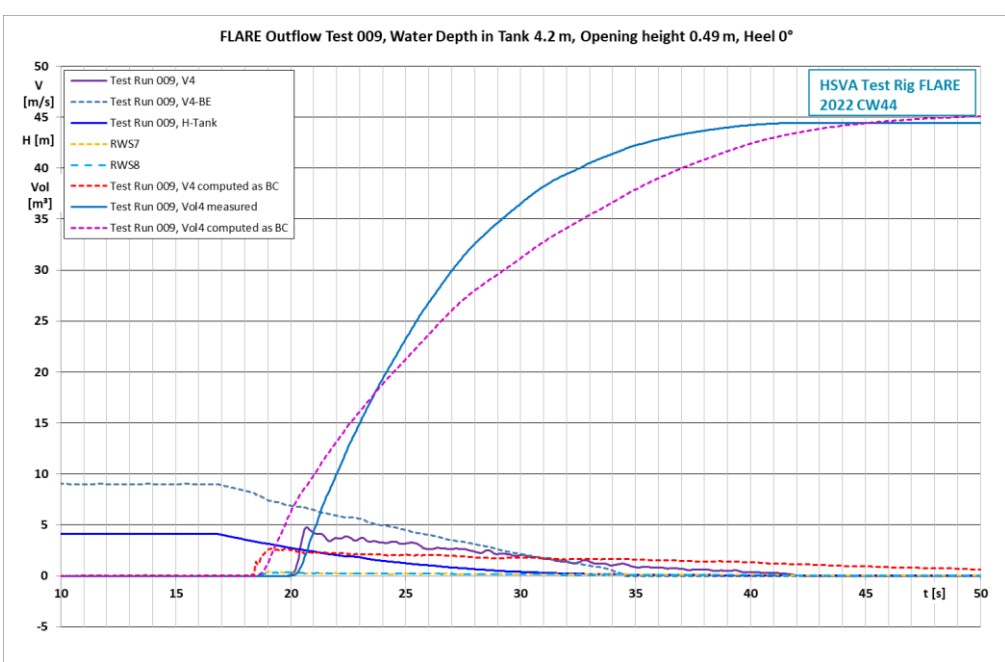

**Figure A5.** Flow speed $V_4$ and water elevations RWS7 and RSW8 at the fourth section $S_4$ for the test run 009 as earlier. In addition, the water discharge volumes $Vol_4$ based on the flow speed $V_4$ computed with BE and based on the measured flow speed $V_4$ are shown. These two curves coincide only in this particular case.

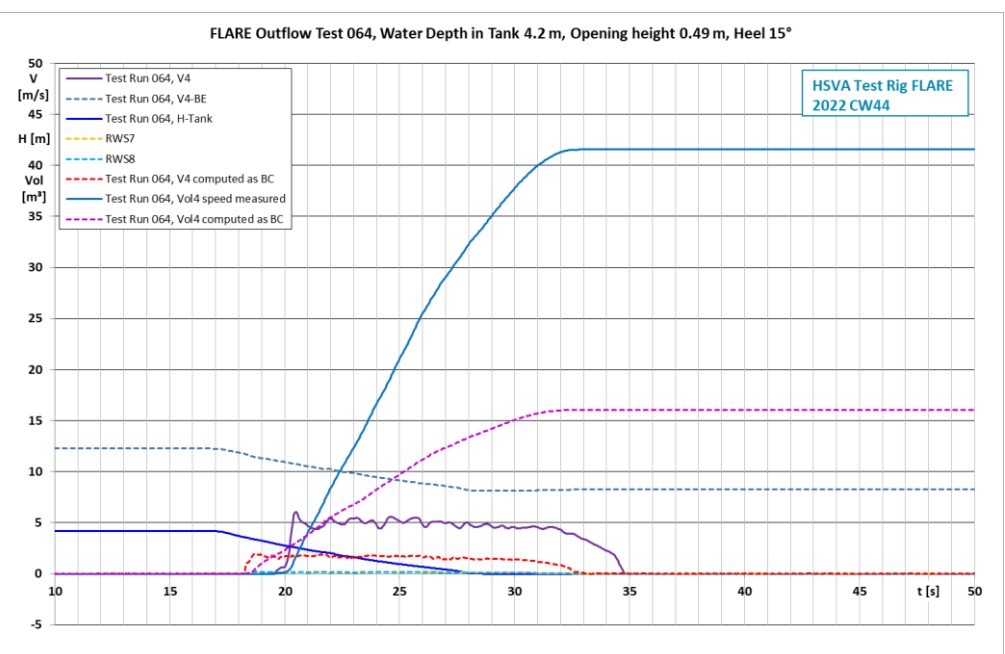

**Figure A6.** Flow speed $V_4$ and water elevations RWS7 and RSW8 at the fourth section $S_4$ for the test run 064. The water discharge volumes $Vol_4$ based on the flow speed $V_4$ computed with BE and based on the measured flow speed $V_4$ are also shown.

Table A1 shows the maximum flow speeds measured at sections $S_1$–$S_4$ for all test conditions. Each maximum value is the average maximum flow speed of five test runs. For easier comparison of the flow speeds, the flow speed in section $S_1$ at zero heeling angle has been given the nominal value of 100 % and the other speeds are scaled accordingly. Note that the nominal water speeds between the different initial water depths in the tank differ.

**Table A1.** Comparison of maximum flow speeds at sections $S_1$–$S_4$ at different initial water depths in the tank and heeling angles of the test rig. For each water depth, the $V_1$ sensor at zero heeling angle has the nominal value of 100%.

| Water Depth [m] | Heeling Angle [°] | Flow Speed at Sections $S_1$–$S_4$ | | | |
|---|---|---|---|---|---|
| | | $V_1$ [%] | $V_2$ [%] | $V_3$ [%] | $V_4$ [%] |
| 0.7 | 0 | 100 | 105 | 94 | 77 |
| 0.7 | 5 | 169 | 162 | 160 | 142 |
| 0.7 | 10 | 197 | 217 | 204 | 193 |
| 0.7 | 15 | 222 | 261 | 139 | 224 |
| 0.7 | 20 | 268 | 300 | 297 | 255 |
| **Water Depth [m]** | **Heeling Angle [°]** | **Flow Speed at Sections $S_1$–$S_4$** | | | |
| | | $V_1$ [%] | $V_2$ [%] | $V_3$ [%] | $V_4$ [%] |
| 2.1 | 0 | 100 | 86 | 85 | 78 |
| 2.1 | 5 | 113 | 112 | 92 | 88 |
| 2.1 | 10 | 118 | 125 | 97 | 103 |
| 2.1 | 15 | 125 | 135 | 117 | 114 |
| 2.1 | 20 | 130 | 144 | 136 | 123 |
| **Water Depth [m]** | **Heeling Angle [°]** | **Flow Speed at Sections $S_1$–$S_4$** | | | |
| | | $V_1$ [%] | $V_2$ [%] | $V_3$ [%] | $V_4$ [%] |
| 4.2 | 0 | 100 | 80 | 75 | 52 |
| 4.2 | 5 | 102 | 95 | 80 | 70 |
| 4.2 | 10 | 105 | 101 | 66 | 78 |
| 4.2 | 15 | 108 | 105 | 91 | 81 |
| 4.2 | 20 | 110 | 107 | 100 | 85 |

Two tendencies are present at all initial water depths in the tank: (1) the maximum (peak) speed reduces as the water flows towards the opening on the right hand side; and (2) the maximum (peak) flow speed increases significantly with the increasing heeling angle. The highest relative increase takes place at the low initial water depth of 0.7 m in the water tank. In Section $S_4$ near the outflow boundary, the outflow speed (255%) with a heeling angle of 20° is ca. 3.5 times the speed with a zero heeling angle (77%).

*Conclusions on the Tests with the Inclined Rig*

**Phase difference:** In the experiments, the inertia of the floodwater mass delayed any change in the flow speed or in the flow direction, whereas the typical boundary condition for inflow/outflow through the damage opening based on the Bernoulli equation does not do this. When the ship is floating in waves, the inflow to and outflow from damaged compartments are continuously changing. In the numerical simulations, the lack of inertia in the Bernoulli model can cause a phase difference in the inflow and outflow through the damage opening in comparison with the experiments. This may influence the excitation of the ship's rolling motion in waves.

**Increase in the net water discharge through the damage opening:** The inertia of the floodwater has a slowing effect on the continuously changing flow speed through the damage opening. The use of the BE for the inflow/outflow through any damage opening in the numerical models for damaged ships can lead to a somewhat too high floodwater discharge through the opening and a too short predicted time to flood or capsize.

**Outflow from inclined decks:** The tests carried out with the rig show that the floodwater on the damaged vehicle deck of a RoPax or on another deck in any ship can develop

considerable outflow speeds, when the ship heels and the deck gets an inclination angle. This speed of the mostly shallow water flow on the inclined deck can be clearly higher than the outflow speed described by the Bernoulli equation as a boundary condition for the damage opening. Thus, the flow speed on the deck should be determined and taken into account in the boundary condition for the damage opening in the numerical simulations. Omitting the outflow speed calculation and using the classical BE as a boundary condition can lead to a too high net accumulation of floodwater on the vehicle deck and thus to a too short time to flood or time to capsize.

**Note**

1.  Maritime Safety Research Centre of the University of Strathclyde, UK.

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
