# Peer review of "On Boundary Conditions for Damage Openings in RoPax-Ship Survivability Computations"

_jmse, doi:10.3390/jmse11030643_

Round 1
Reviewer 1 Report
Dear author,
I would like to thank you for your efforts in preparing this article.
The paper is concerned with floodwater modeling through damaged ship side. The paper is well written, concise and to the point citing the most recent and relevant literature. I am in favour of publishing this paper. There are only few minor suggestions to the author:
1. Section 3.1. Could the authors be more specific what means free-drifting mode and the soft-moored ship model tests. Some details about those two mades and differences would be beneficial to readers not familiar with the terms.
2. Consider improving th equality of figures, especially fig 8-10. The legend text is bit foggy and in fig 10 it is difficult to discern colors (especially the blue variants)
I hope that these minor tweaks improve already a nicely written paper.
Kind regards,
Your reviewer
Author Response
Thank you for your review. I have improved the points mentioned by you.
Reviewer 2 Report
The author presents a convincing case for adding more complex equations for numerical modelling of the the flow on and off the deck of a damaged ro-pax vessel. There was clearly a lot of thought went into the design of the experiments, and the modifications to an existing computer code. Overall, the paper is well written and summarizes an extensive body of work.
One thing that is not discussed is the change, if any in computation time caused by adding the extra steps. Also, the discussion section, as presented, should perhaps be in the introduction or as part of the motivation for the study. The discussion should focus on the results of the study, and if it is a benefit, to compare those to other published work.
In the course of the review, I identified some minor spelling or grammatical errors. These are;
Line 149 wave trough
Line 161 we assume the flow to be unsteady
Figure 4 caption, flood water flowing out
Line 247, it is difficult to make reliable measurements (more concise...)
Line 335, we only have the author's word that the method is more reliable at this point in the paper. Perhaps it should be described as a more complete formulation?
Author Response
Thank you for your review and good suggestions. I have added /corrected according to your first five suggestions. The last one ( Line 335 ..) I cannot really improve.
Reviewer 3 Report
The paper gives discussion on Boundary Conditions for Damage Openings in RoPax-Ship Survivability Computations and discusses the potential of alternative to Bernoulli models in for use in rapid simulation models. The content and objectives of the article are sound especially from an engineering practitioner's perspective. The language is excellent and the conclusions satisfactory. In my opinion the article can be accepted.
Author Response
Thank you for your review and comments. Based on all reviewers comments I got, the paper has still become better.